# Cretaceous Connections Among Camel Cricket Lineages in the Himalaya Revealed Through Fossil-Calibrated Mitogenomic Phylogenetics

**DOI:** 10.3390/insects16070670

**Published:** 2025-06-27

**Authors:** Cheten Dorji, Mary Morgan-Richards, Steven A. Trewick

**Affiliations:** 1College of Natural Resources, Royal University of Bhutan, Thimphu P.O. Box 11001, Bhutan; 2Wildlife & Ecology, School of Food Technology and Natural Sciences, Massey University, Manawatū, Palmerston North 4442, New Zealand; m.morgan-richards@massey.ac.nz

**Keywords:** Rhaphidophoridae, molecular clock, phylogeny, camel cricket, cave cricket, sympatry

## Abstract

The flightless camel crickets are one of the oldest living lineages of Orthoptera, believed to have originated around 250 million years ago. In this paper we infer the timing of their radiation using DNA sequences from whole mitochondrial genomes of 20 camel crickets, with a focus on the neglected Rhaphidophorinae and Aemodogryllinae groups. To determine whether the taxonomic groups share a single common ancestor, we combined new DNA sequences from camel crickets from Bhutan with published genetic data. Our phylogenetic tree supports the monophyly of most of the genera sampled but supports the reinstatement of *Gymnaeta* Adelung, which forms a lineage sister to the group comprising *Diestrammena*, *Tachycines*, *Gymnaetoides*, *Homotachycines*, and *Pseudotachycines*. Based on our fossil-calibrated molecular clock phylogeny, the common ancestor of camel crickets was estimated to have lived in the Early Jurassic when the supercontinents were still connected. We estimate that the most recent common ancestor of Aemodogryllinae and Rhaphidophorinae lived about 137 million years ago, well before America and Asia were connected by the Bering Land Bridge. Thus, we find little evidence to suggest that continental drift explains the current distribution of this wingless orthopteran family.

## 1. Introduction

The Rhaphidophoridae, commonly known by the names camel crickets, cave crickets, and cave wētā, are among the least-studied groups within the Orthoptera [1]. The family is diverse, encompassing 912 extant described species and 93 genera [2] distributed around the world. Although all species of Rhaphidophoridae are flightless, there are numerous examples living on geologically young oceanic islands [3,4,5] resulting from colonization, which demonstrates their excellent dispersal potential [6]. They are characterized by an arched profile; long antennae; and usually long legs, an absence of wings and tympanum, and limited pigmentation (Figure 1). They are nocturnal and associated with humid environments in temperate and tropical wet forests and sometimes in caves.

The Rhaphidophoridae are classified into nine extant and one extinct subfamily, each restricted to a distinct geographic region (Figure 1): the Troglophilinae and Dolichopodainae are found in Mediterranean Europe; the Rhaphidophorinae, Aemodogryllinae, and Anoplophilinae in Asia, with the latter being limited to the Korean Peninsula and Japan; the Macropathinae in the Southern Hemisphere; and the Ceuthophilinae, Gammarotettiginae, and Tropidischiinae exclusively in North America. Although representation in molecular phylogenetic analyses is sometimes sparse, the monophyly of seven subfamilies is concordant with their distinct geographical distributions [3,7]. The geographic partitioning of relatively deep lineages implies long regional persistence in separate regions, but this pattern does not necessarily reveal paleogeographic processes due to the unknown influence of extinction [8].

The Rhaphidophoridae are regarded as a sister to the majority of ensiferan diversity and are thought to have originated between the Carboniferous and Late Jurassic periods [9,10,11]. Vicariant explanations of the global distribution of the Rhaphidophoridae rather than long-distance dispersal have historically been favored, primarily due to the absence of wings in all species [12,13,14], and phylogenetic trees are broadly consistent with this [3,7,15,16,17,18]. While Northern Hemisphere subfamilies are each associated with one part of a continent, the Macropathinae are distributed across multiple continents, including South Africa, South America, Australia, New Zealand, and several subantarctic islands. Data-rich phylogenetic analyses of this subfamily show that rhaphidophorids have successfully dispersed long-distances across inhospitable habitats [6], and it is widely recognized that dispersal is the only plausible explanation for the current spatial arrangement of many endemic species on young oceanic islands [19,20,21,22]. Sister to the Macropathinae are the Northern Hemisphere subfamilies the Aemodogryllinae and Rhaphidophorinae, which have their nexus in Southeast Asia, a geologically complex region of continent and islands. How did these distinctive lineages come to occupy the same region? One suggestion is that their common ancestor colonized East Asia in the later Cretaceous via the Bering Land Bridge [7].

The timing of lineage origins and the relevance of landscape processes in the evolution of the Orthoptera and other insects often relies on the application of paleogeographic events for phylogenetic dating [3,23,24], but this is problematic since it is founded on an assumption of the very question to be decided [25]; it assumes a role of landscape changes in lineage formation, which results in logical circularity [19,26,27]. However, the scarcity of relevant verified fossils inhibits the use of this alternative to calibration [28].

The earliest fossils assigned to the Rhaphidophoridae are from Oligocene Baltic amber [29]. *Protroglophilus* and *Prorhaphidophora* [29,30] represent lineages of uncertain affinity and are not definitively stem lineages of the family or extant subfamilies and are therefore unsuitable for molecular phylogeny that seeks to estimate the divergence of the major phylogenetic linages. Uncertainty in the systematic placement of other extinct taxa in the Tettigoniidea, such as *Aenigmaraphidophora mouniri* [31], prevents their use. Consequently, previous molecular phylogenetic studies on the Rhaphidophoridae used inferred paleogeographic events to calibrate the most recent common ancestor of the group [3,15,32]. To overcome the limited availability of fossils, ‘secondary fossil’ calibration relies on fossil-constrained calibration of a different taxon set that overlaps the group of interest, with node dates being transferred for molecular clock analysis of the focal group [7,16].

Camel crickets from the Asian continent form a major part of the global Rhaphidophoridae diversity, yet their phylogenetic relationships have been neglected. For *Tachycines* and *Diestrammena* within the subfamily Aemodogryllinae, classification of species is challenging due to the large number of species, their high degree of morphological similarity, and the lack of diagnostic characters. The genus *Tachycines* was established with *T. asynamorus* by separation from *Diestrammena* based on a difference in the number of spines on the hind tibia [33]. Furukawa, in 1933, did not accept the evidence for the division of *Diestrammena* and *Tachycines* proposed by Adelung and preferred *D.* (*Tachycines*) [34]. Karny (1934) merged *Diestrammena* with *Gymnaeta* Adelung as subgenera of *Tachycines*; hence *Tachycines* (*Diestrammena*) and *T.* (*Gymnaeta*) [13]. Gorochov and Storozhenko (1992) transferred the subgenus *Gymnaeta* from the genus *Tachycines* to *Diestrammena* [35]. Six years later, Gorochov (1998), using male genitalia, proposed four subgenera of *Diestrammena*: *D.* (*Diestrammena*), *D.* (*Aemodogryllus*), *D.* (*Tachycines*), and *D.* (*Gymnaeta*) [36]. Despite the complicated and convoluted taxonomic history of these cricket species, their systematics has not been examined using molecular phylogenetics.

We sampled taxa representing five subfamilies to determine the evolutionary relationships of the Rhaphidophoridae, including systematic analysis of neglected Asian subfamilies, the Aemodogryllinae and Rhaphidophorinae. We constructed phylogenetic hypotheses from mitogenomic data and inferred the timing of the diversification of major lineages using fossil-calibrated molecular clock analysis to shed light on the biogeographic history of the family. Explicitly we examined the proposal that the most recent common ancestor of the Aemodogryllinae and Rhaphidophorinae existed in the Late Cretaceous, facilitating dispersal to North America and East Asia from Beringia [7].

## 2. Materials and Methods

### 2.1. Taxon Sampling

In this study, we considered species representing six subfamilies of the Rhaphidophoridae from three major Northern Hemisphere geographical regions (North America, Europe, and Asia) and the Southern Hemisphere. Two datasets were generated. The larger taxon dataset (n = 117 species) was compiled to test the current generic classification system within the Aemodogryllinae and Rhaphidophorinae by incorporating data from as many species representatives as possible. We supplemented data available from previous phylogenetic studies of the Rhaphidophoridae (Aemodogryllinae and Rhaphidophorinae) from Asia (e.g., [16,37]) with sampling from Bhutan, Oceania, and North America. Samples from Bhutan were collected during April 2022 under the authorization of research permit number 7008186946225A7A8C6D0B granted by the Ugyen Wangchuck Institute for Forestry Research and Training (UWIFoRT), Bhutan. Subsequent transfer to Massey University was conducted in accordance with a Material Transfer Agreement, reference number NBC/BRD/7/2022-2023/197. All samples were deposited in the Phoenix collection at Massey University, Palmerston North (MPN). We used DNA sequences (12S, 16S, COI, 28S, and 18S) from GenBank and the Barcode of Life Data System (BOLD) in conjunction with additional genetic data extracted from skim sequencing (Appendix A). This dataset will be referred to as the 117-taxa set.

For molecular clock analyses we compiled a dataset using the 13 protein-coding genes extracted from whole mitochondrial assemblies of 20 Rhaphidophoridae and 2 outgroup taxa (Table 1). This included representatives of diversity in Southern Hemisphere Macropathinae that constitute a sister to the clade that includes the Aemodogryllinae and Rhaphidophorinae in Asia and near Oceania. We generated new mitogenomic data for species in the subfamilies Rhaphidophorinae and Aemodogryllinae from Bhutan and the Solomon Islands and Ceuthophilinae from the United States of America. The Rhaphidophorinae and Aemodogryllinae were identified based on the descriptions provided in [35,38,39,40,41]. The Rhaphidophorinae specimen from Vangunu Island, Solomon Islands, was identified as belonging to the genus *Stonychophora* Karny upon comparing descriptions and high-resolution images of the type specimen *Stonychophora salomonensis* Willemse described from Aola, Solomon Islands [13,42]. We identified the specimen belonging to the Ceuthophilinae as a member of the genus *Ceuthophilus* Scudder [43].

### 2.2. DNA Sequencing

We assembled entire mitochondrial genomes from DNA extractions without the need for PCR using skim sequencing techniques. For this, we carried out extraction of DNA from leg muscle using a salting-out protocol [22,44]. Genomic DNA samples were paired-end-sequenced through massive parallel, high-throughput sequencing on an Illumina HiSeq 2500 by Macrogen (Seoul, Republic of Korea) following fragmentation and indexing using the Illumina (San Diego, CA, USA) TruSeq Nano DNA kit. The resulting DNA reads with a mean fragment size of 150 base pairs (bp) were sorted and iteratively assembled in Geneious Prime v2022.2.2 [45]. Initial mapping reads from each sample used published annotated mtDNA genome data from GenBank under strict sensitivity settings to allow mapping with minimal gaps and ambiguity, generating novel consensus sequences [46,47]. Partial assemblies were remapped to close gaps and the resulting draft assembly remap with raw sequence reads until all the alignment gaps were filled by extension with the new sequence data and ambiguities were resolved. Consensus sequences were checked for ambiguity and reading-frame anomalies and annotated by comparison with published examples, with submission to the MITOS server [48]. Reading frames, amino acid translation, and secondary structures were examined for protein-coding, ribosomal RNA (rRNA), and transfer RNA (tRNA) genes, respectively. Each of the 13 protein-coding genes were extracted and aligned separately using ClustalW aligner (default settings) before they were concatenated to give a final dataset 11,124 nucleotides in length.

**Table 1 insects-16-00670-t001:** Rhaphidophoridae and sister Prophalangopsidae used in phylogenetic analysis of Asian subfamilies. DNA sequence data consist of 13 mitochondrial protein-coding genes, 22 tRNAs, and 2 rRNAs (14,800 bp). Taxa newly sequenced in this study are in **bold**, and their specimen codes are linked with specimens kept in the Phoenix Collection at Massey University, Palmerston North (MPN). * The names associated with GenBank accessions for *Diestrammena asynamora* and *Diestramima* sp. we hereafter refer to as *Tachycines asynamorus* and *Diestramima intermedia*, respectively, in accordance with [16,49].

Family	Subfamily	Taxa	Country	Specimen Code	Accession No.	Author
Rhaphidophoridae	Aemodogryllinae	*Diestramima matermagna*	Pema Gatshel, Bhutan	MPN_CW5536	**OR896621**	**This study**
		*Diestramima tsongkhapa*	Trongsa, Bhutan	MPN_CW5525	**OR896622**	**This study**
		*Diestramima intermedia* *	China		KX057718	[50]
		*Diestramima tibetensis*	China		KX057740	[50]
		*Diestrammena* sp.	China		MT849270	[51]
		*Diestrammena japanica*	Japan		MK347245	[52]
		*Tachycines asynamorus* *	China		KX057726	[50]
		*Tachycines shuangcha*	China		OM993275	[53]
		*Tachycines zorzini*	China	MW322826	NC_057442	[54]
	Ceuthophilinae	*Ceuthophilus* sp.	Moab Desert, USA	MPN_CW4347	**OR880641**	**This study**
	Rhaphidophorinae	*Stonychophora* sp.	Vangunu, Solomon Is.	MPN_ORT15	**OR896624**	**This study**
		*Rhaphidophora quadrispina*	China		OL450400	[55]
		*Rhaphidophora bicuspis*	Thimphu, Bhutan	MPN_CW5529	**OR896623**	**This study**
		*Rhaphidophora bhutanensis*	Pema Gatshel, Bhutan	MPN_CW5483	**OR896625**	**This study**
		*Rhaphidophora bilobata*	Trongsa, Bhutan	MPN_CW5545	**OR896626**	**This study**
	Troglophilinae	*Troglophilus neglectus*	Brje pri Kombu, Slovenia		EU938374	[56]
	Macropathinae	*Macropathus* sp.	Waitomo, NZ	MPN_CW109	OR520204	[6]
		*Talitropsis sedilotti*	Hawkes Bay, NZ	MPN_CW1830	OR551721	[6]
		*Parvotettix domesticus*	Taronga, Tasmania	MPN_CW736	OR551716	[6]
		*Spelaeiacris monslamiensis*	Hex River, South Africa	MPN_CW3801	OR551731	[6]
Prophalangopsidae	Cyphoderrinae	*Cyphoderris monstrosa*			KM657332	[1]
	Prophalangopsinae	*Tarragoilus diuturnus*			NC_021397	[50]

### 2.3. Phylogenetic Analysis

For analysis of the 117-taxa set, we concatenated partial DNA sequences of three mitochondrial genes (12S, 16S, and COI) and two nuclear ribosomal genes (28S and 18S) to provide a total alignment length of 3314 bp. For phylogenetic inferences the data were partitioned based on gene type (ribosomal or protein-coding) and on codon position (for COI), creating seven partitions.

Phylogenetic reconstruction of five subfamilies within the Rhaphidophoridae used data from whole mitochondrial genomes. For these we extracted and concatenated the protein-coding genes and excluded the tRNA and rRNA genes based on the findings from previous studies on Orthoptera that found that the use of only protein-coding genes resulted in the most stable analyses and that inclusion of RNAs did not improve tree topology [46,57,58]. We explored the influence on topology and node support of different partition models using the Maximum Likelihood (ML) method on both nucleotides and amino acid sequence alignments. The final dataset for 20 taxa consisted of concatenated data for 13 protein-coding genes, and removing stop codons from each gene and trimming to equal alignment lengths yielded 13 partitions based on genes and 52 partitions based on genes plus codon positions.

For both datasets, we used Partition Finder 2 [59] implemented in IQ-TREE 2 v2.2.0 [60] to identify the optimal partition scheme and the most suitable models of DNA evolution under the Bayesian information criterion (Appendix A). The best partitioning schemes were used for reconstructions of phylogenetic relationships within the family. The ML analysis was performed in IQ-TREE 2 considering invariable sites, Gamma rate heterogeneity, and resampling strategy [61]. Ultrafast Bootstrap [62] and Sh-aLRT support values [63] were calculated with 1000 replicates.

### 2.4. Divergence Time Estimate Analysis

We inferred the timing of lineage splitting among the rhaphidophorid subfamilies using BEAST2 v2.7.4 [64]. Since the only fossil record of Rhaphidophoridae is uninformative for dating the earliest splits, recent studies have relied on estimates of time of divergence using secondary fossil calibration points from a phylogenomic study of Orthoptera (e.g., [7,16]). The reliability of node ages of subfamily linages within the Rhaphidophoridae is uncertain due to under-representation of taxa (only 3 Rhaphidophoridae included in [65]), which may result in significant skewing in molecular clock inferences. We also used a secondary fossil calibration but in conjunction with a fossil calibration constraining the sister lineage of Rhaphidophoridae. We calibrated our phylogeny using the fossils of Prophalangopsidae, which are considered to be close relatives of the rhaphidophorid family [1,65]. Positioning a calibration within the target group of analysis is essential for good molecular clock inference; therefore, we positioned a secondary fossil calibration to constrain the node age of the common ancestor of Macropathinae and (Aemodogryllinae + Rhaphidophorinae). Our secondary fossil calibration within the Rhaphidophoridae is based on an analysis that included the same Prophalangopsidae fossil calibration but with much wider outgroup sampling (nine taxa representing Anostostomatidae, Stenopelmatidae, and Tettigoniidae) and included a well-dated geological constraint within their ingroup using endemic *Talitropsis* species of the Chatham Islands [6]. We constructed our phylogeny using mitochondrial genomes. We examined the effects on divergence times, model convergence, and Effective Sampling Size (ESS) scores of applying different distribution priors (normal, log-normal, and exponential) [28]. We found that a normal distribution prior on both the fossils outside the rhaphidophorids and on a secondary node constraining Macropathinae + Aemodogryllinae + Rhaphidophorinae gave the optimal result, and these were used for the final analysis (Table 2).

Fossils belonging to the genus *Aboilus* Martynov (201.3–157.3 million years ago [Mya]) are the oldest definitive Prophalangopsidae known [66,67]. These are well-recognized fossils previously used in divergence dating analysis of Orthoptera [1,50]. *Aboilus* consists of 20 extinct taxa [2], with fossils from the Jurassic and Cretaceous. We constrained two commonly used extant species of Prophalangopsidae in our molecular analyses (represented by *Cyphoderris monstrosa* Uhler and *Tarragoilus diuturnus* Gorochov) under a normal distribution prior with the minimum fossil age as a minimum soft bound (157.3 Mya) and 192 Mya as a maximum soft bound.As a secondary fossil calibration point, we constrained our analysis using a recent phylogenomic study of Southern Hemisphere rhaphidophorids calibrated with a Prophalangopsidae fossil (*Aboilus*) and a recent geological constraint [6]. We used 160 Mya as a mean age, with 95% Highest Posterior Density (HPD) values as soft minimum and maximum bounds at the node of Macropathinae and Aemodogryllinae + Rhaphidophorinae.

The BEAST input file was generated using BEAUti2 v2.7.5 [68] by implementing parameters for molecular clock models, trees, and fossil priors. As in the ML phylogeny, we used optimal partitioning by genes in the concatenated protein-coding mitochondrial DNA dataset by linking clock and tree models for three final substitution models. We implemented a GTR model with non-standard substitution models by allowing parameter linking. Fossil-calibrated analyses were run under an uncorrelated optimized relaxed clock model with the birth–death process as a tree prior.

For comparison we also calibrated dating analyses of the same DNA dataset using two node age estimates derived from a previous study that relied on the paleogeographic inference of the timing of ancient land distributions (referred to here as ‘landscape’ calibration). This approach makes the assumption that continental drift was the primary influence on lineage splitting in the Rhaphidophoridae [3]. Thus, the split between the North American Ceuthophilinae and the European Troglophilinae was set at 68.4 Mya with 95% HPD bounds of 46.3–99.3, and the ancestor of Aemodogryllinae and Rhaphidophorinae with Macropathinae was set to 117 Mya with 95% HPD bounds of 105–130. We enforced similar settings as above in BEAUti2 using a relaxed molecular clock and the birth–death process as a tree prior. We used 95% HPD values of these node ages as soft minimum and maximum bounds in a normal prior distribution around the secondary landscape calibration points.

**Table 2 insects-16-00670-t002:** Internal node estimates (most recent common ancestors (MRCAs)) of subfamilies of Northern Hemisphere camel crickets (Rhaphidophoridae). Node ages inferred from primary calibration using an orthopteran fossil and secondary fossil-calibrated node ages (above) and (below) from secondary landscape calibration based on estimated inferred paleogeographic history [3,69]. Median ages and 95% Highest Posterior Density (HPD) credibility intervals are in millions of years ago (Mya). Abbreviation used for subfamilies: Ceuthophilinae (Ceu); Troglophilinae (Tro); Macropathinae (Mac); Rhaphidophorinae (Rha); and Aemodogryllinae (Aem). Prior distribution models for fossil (F) and secondary (S) calibrations were U = Uniform, LN = Log-Normal, N = Normal, and E = Exponential. Number of generations (N. gen.) in millions. Effective sampling sizes are provided for posteriors, tree likelihoods, and priors after removing 10–40% as burn-in.

**BEAST RUN**	**Priors F/S**	**N. Gen.**	**Posterior**	**Tree Likelihood**	**Prior**	**Rhaphs. MRCA**	**95% HPD**	**Ceu and Tro MRCA**	**95% HPD**	**Rha and Aem MRCA**	**95%HPD**	**Fossil Treatment: *Aboilus* (201–157 Mya)**	**Sec. Treatment: Mac/Aem + Rha Node Age (140–180 Mya)**
1	LN/N	10	411	327	368	188.7	166.6; 210.7	153.4	131; 178.7	137.6	121.5; 154.9	Hard min = 157; mean =172; 97.5% max = 222	2.5% min = 140; mean = 160; 97.5 max = 180
2	N/N	10	1109	846	1819	188.8	165.7; 211.3	153.6	129.8; 178	138.1	120.7; 154.9	2.5% min = 152; mean = 172; 97.5% max = 192	2.5% min = 140; mean = 160; 97.5 max = 180
3	E/E	10	1019	775	1234	188.7	159.2; 226.8	153.7	124.7; 189.7	137.6	115.2; 166.1	Hard min = 157; mean = 172; 97.5% max = 212	Hard min = 140; mean = 160; 97.5% max = 214
4	LN/E	10	945	675	1173	188.6	156.4; 227.2	153.6	126.1; 189.7	135.3	113.2; 170.6	Hard min = 157; mean = 172; 97.5% max = 222	Hard min = 140; mean = 160; 97.5% max = 214
5	N/N	100	10278	7661	20,075	188.7	166.7; 211.9	153.6	130; 178.5	138.1	121.1;155	2.5% min = 152; mean = 172; 97.5% max = 192	2.5% min = 140; mean = 160 97.5 max = 180
**BEAST RUN**	**Prior S/S**	**N. Gen.**	**Posterior**	**Tree Likelihood**	**Prior**	**Rhaphs. MRCA**	**95% HPD**	**Ceu and Tro MRCA**	**95% HPD**	**Rha and Aem MRCA**	**95% HPD**	**SEC. Node Age Treatment: Ceu and Tro (99.3–46.3 Mya)**	**Sec. Node Age Treatment: Mac/Aem + Rha Node Age (130–105 Mya)**
1	N	10	707	663	1245	109.5	98.0; 122.4	69.9	65.8; 73.6	88.5	77.6; 100.8	2.5% min = 48.9; mean = 68.5; 97.5% max = 88	2.5% min = 105; mean = 117; 97.5% max = 129
2	E	10	635	511.1	1266	113	105.3; 127.7	85.4	69.8; 102.2	90.4	81.4; 103.8	Hard min = 56; mean = 68.5; 97.5% max = 102	Hard min = 105; mean = 117; 97.5% max = 149
3	U	10	776	626.9	1388.3	121	106.6; 136.5	94.5	76.2; 115.9	95.5	84.1; 108.5	Bounds: min = 46; mean = 72.5; max = 99	Bounds: min = 105; mean = 118; max = 130
4	N	100	8860	8032	10,664	109.8	97.7; 122.1	69.8	66; 73.6	88.3	77.1; 100.4	2.5% min = 48.9; mean = 68.5; 97.5% max = 88	2.5% min = 105; mean = 117; 97.5% max = 129

For both calibration approaches we enforced monophyly constraints on each subfamily for consistency in topologies of molecular clock dating trees with the ML tree topology from 13 protein-coding genes. Initially, to assess the convergence of models using different distribution priors, we conducted BEAST2 analysis of 10 million generations. After considering analysis convergence and evaluation of Effective Sampling Sizes, final BEAST2 runs of 100 million generations with sampling every 1000 generations were performed. Tracer v1.7 [70] was used to inspect the results, and ESS statistics were investigated to check the fit of the models. We considered ESS values greater than 250 sufficient for the analyses to be informative after discarding 10% of the run as burn-in. Maximum clade credibility trees with median heights were generated in TreeAnnotator v2.7.1 [71] after removing fossils and were visualized in FigTree v1.4.4 [70].

## 3. Results

### 3.1. Systematics

The ML tree for the set of 117-taxa resulted in six well-supported clades congruent with recognized subfamilies of the Rhaphidophoridae. All the clades were supported by high statistical bootstrap values > 90 (Figure 2). Although diverse in terms of body size and color patterns, most rhaphidophorid genera are recognized by a few traits that typically include apical spines on legs and features of male terminalia. Nevertheless, our phylogenetic hypothesis was mostly concordant with the current taxonomy. Within the subfamily Aemodogryllinae, four of the six genera sampled formed separate monophyletic clades. The notable exception to this was the placement of six taxa in the genera *Diestrammena* Brunner von Wattenwyl and *Tachycines* Adelung in three separate paraphyletic clades: (a) *Tachycines asynamorus* (Adelung) with *Tachycines shuangcha* Feng, Huang & Luo having a close relationship to *Gymnaetoides* Qin, Liu & Li; (b) *Diestrammena japanica* Blatchley with *Diestrammena fengyangshanica*; and (c) *Tachycines zorzini* (Rampini & Di Russo) with *Diestrammena* sp. (Figure 2).

### 3.2. Rhaphidophoridae Mitogenomic Data

New mitochondrial genomes were assembled and annotated for seven rhaphidophorids. For all sampled taxa (Table 1), gene order was typical of mitochondrial genomes already documented for Rhaphidophoridae, comprising the expected complement of 13 protein-coding genes, 22 tRNAs, 2 rRNAs, and a repeat region (putative control region) (Appendix A) [6]. The final length of 13 concatenated protein-coding genes varied from 11,208 bp (MPN_CW4347) to 11,231 bp (MPN_CW5536 and MPN_CW5525), with truncated AT stop codons in COII, ND4, and ND5 in some sampled taxa (Appendix A). Assembled mitochondrial genomes, excluding the putative control region that lies between the small rRNA and tRNAs adjacent to ND2, ranged in size from 14,718 bp (MPN_ORT15) to 14,885 bp (MPN_CW5529).

### 3.3. Phylogenetic Analysis and Time Calibration

Phylogenetic analysis of the concatenated alignment of 13 mitochondrial protein-coding genes (11,124 bp) yielded a resolved tree topology with high node bootstrap support (99 or 100%) (Appendix A). Topologies from different partition and model schemes on nucleotide and amino acid alignments were congruent and without significant differences in node support values, with a gene concordance factor > 45 (Appendix A). The topology was found to be consistent with the current classification of rhaphidophorid subfamilies. Despite the smaller number of taxa representing the subfamily Aemodogryllinae, the paraphyletic nature of the genera *Diestrammena* and *Tachycines* was consistent with our findings from the analysis of a combination of partial gene sequences for 117-taxa. *Stonychophora* from the Vangunu Island, Solomon Islands (MPN_ORT15), were grouped within the Asian subfamily Rhaphidophorinae but on a long branch.

Following standard practice [69,72,73], initial fossil calibration and secondary landscape calibration analyses were performed using 10 million generations for each under different combinations of prior distributions to access the effects on convergence with reference to Effective Sampling Size (ESS) statistics examined in Tracer. During this process, it was recognized that all the analyses started to converge between 1 and 4 million generations, with stable node ages and good ESS values (>250) for all the priors (after removing 10–40% as burn-in), with the exception of secondary landscape calibration that showed varying node ages and 95% HPD intervals. Analysis using a combination of log-normal and exponential priors yielded a wide range of upper and lower bounds for all node ages. For the final analysis, normal priors for both the calibration points (fossil and secondary fossil calibration) were found to be optimal based on the evaluation of ESS statistics (Table 2, Beast run 5). For secondary landscape calibration, uniform and exponential distribution priors produced slightly higher median ages compared to results with a normal prior and very wide ranges of minimum and maximum 95% HPD intervals for the node ages of Troglophilinae and Ceuthophilinae. Final analysis of secondary landscape calibration was therefore performed using normal prior distributions (Table 2, Beast run 4).

Our fossil-calibrated molecular clock analysis estimated the origin of Northern Hemisphere Rhaphidophoridae subfamilies during the Jurassic and Early Cretaceous periods of the Mesozoic Era (Figure 3). The earliest common ancestor of the Rhaphidophoridae we sampled is estimated to have lived during the Early Jurassic, 188.7 Mya (Table 2). We estimated that the split between the North American Ceuthophilinae and Mediterranean Troglophilinae (~153 Mya) and Asiatic lineages from Southern Hemisphere Macropathinae (~164 Mya) also occurred during the Mesozoic. The Aemodogryllinae and Rhaphidophorinae diverged during the Cretaceous (~138 Mya). The most recent common ancestor within the Rhaphidophorinae diverged at about 105 Mya in the Early Cretaceous, represented by the node between *Stonychophora* from Vangunu Island, Solomon Islands (MPN_ORT15), and the remaining Asian taxa diverged throughout the Cenozoic Era. The common ancestor of the sampled Aemodogryllinae taxa was estimated at 110 Mya.

In contrast to the fossil-calibrated analysis, our secondary landscape calibration of the phylogeny based on inferences from landscape history suggests significantly younger node ages with narrower credibility intervals (Table 2). The estimation of the most recent common ancestor for Rhaphidophoridae indicates a divergence that occurred approximately 110 Mya during the Early Cretaceous. This divergence was followed by the separation of North American and Mediterranean taxa around 70 Mya and the divergence of Asiatic taxa around 108 Mya from the Macropathinae. Within the Asiatic lineages, the common ancestor of the Aemodogryllinae and Rhaphidophorinae lived about 88 Mya. The divergence of the most recent common ancestor of the Pacific taxon and the Asian taxa was estimated to have occurred at 67.8 Mya, while the divergence of the Aemodogryllinae was estimated at 70.5 Mya (Appendix A).

## 4. Discussion

### 4.1. Systematics

Our phylogenetic analysis of 117 Rhaphidophoridae found an excellent concordance between the taxonomy and evolutionary relationships, despite morphological conservation within this family. The placement of only six of the included taxa within *Diestrammena* and *Tachycines* constitutes an exception to the general observation of monophyletic genera. There is inconsistency between the phylogenetic relationships among six *Diestrammena* and *Tachycines* taxa inferred from DNA sequence data with the current classification scheme of two subgenera based on morphology [13,49]. Discrepancies in the assignment of subgenera have also been reported for other genera (e.g., *Diestramima* Storozhenko) in the same subfamily [16]. Adelung (1902) [33] erected the genus *Tachycines*. Since then, the taxonomic placement of taxa associated with *Tachycines* Adelung and *Diestrammena* Brunner von Wattenwyl and with *Gymnaeta* Adelung has been repeatedly modified with the use of the additional subgenus rank [13,34,35,49,74]. The lack of monophyly identified by our analysis of molecular data suggests that a third genus needs to be recognized to resolve the current inconsistencies (Figure 2, clades a, b, and c). This can be achieved by resurrection of the genus *Gymnaeta* Adelung and by revising *Tachycines zorzini* (Rampini & Di Russo) as *Gymnaeta zorzini* and *Diestrammena* sp. (GenBank accession: MT849270) as *Gymnaeta* sp., while maintaining the monophyly of *Tachycines* and *Diestrammena*.

### 4.2. Lineage Age and Origin

Some of the uncertainties of molecular clock analysis, including substitution rate variation, are integrated into molecular dating within the Bayesian framework [75], but the scarcity of appropriate fossil material and the potential for uncertain interpretation of their systematic relationships is inevitable. We showed that the prior distribution models (normal, log-normal, or exponential) used for our calibrations of molecular clock analyses did not significantly impact our estimates of the timing of common ancestors. However, our fossil calibration (in the outgroup) and secondary fossil calibration node (in the ingroup) are not entirely independent of one another and therefore might result in the appearance of greater accuracy than is realistic in our molecular clock analysis. The universal uncertainties from incomplete sampling that results from both extinction and lack of specimens for DNA sequencing must also be considered [26,76]. Our analysis included new samples representing the subfamilies Aemodogryllinae and Rhaphidophorinae from a wider geographic range than previously considered, including the Kingdom of Bhutan in the Himalaya. Better representation of the diversity within these lineages should improve our estimates of the age of their most recent common ancestor. However, the poor sampling from the Pacific Islands and the lack of samples from India, where divergent lineages may exist, remain a major challenge. Therefore, we need to treat all date estimates as hypotheses that require further testing.

If the calibration of our data-rich molecular clock analysis is realistic, then the orthopteran family of camel crickets (Rhaphidophoridae) had a common ancestor before Pangea broke into the supercontinents of Gondwana and Laurasia (~180 Mya). The Jurassic was a major period of continental movement and would appear to have coincided with the divergence of the Mediterranean and North American subfamilies from the Asian and southern subfamilies. However, the confidence intervals around divergence estimates based on fossil calibrations are such that it will be difficult to test vicariant hypotheses based on events that took place so long ago. If the orthopteran fossils from Triassic and Jurassic deposits used for time calibration have been correctly identified, then the age of the rhaphidophorid family is much older than inferred based on an assumption of landscape vicariance [3] and calibration based on secondary constraints derived from a phylogenomic study of Orthoptera [7]. Specifically, the most recent common ancestor of the European lineages (Troglophilinae and Dolichopodainae) and the North American lineage (Ceuthophilinae) seems to have existed before the supercontinent Laurasia broke up to give rise to the basal rocks of Europe, North America, and much of Asia.

Divergence of the Rhaphidophoridae family from a common ancestor during the Late Triassic and Jurassic periods is compatible with the divergence dates of Rhaphidophorids obtained from fossil-calibrated phylogeny studies of Ensifera [1,6,16,65]. The increase in subfamily diversity of Rhaphidophoridae is also consistent with the Jurassic-to-Early Cretaceous peak in insect diversity [77,78], which is in part explained by a warmer climate and the Early Cretaceous radiation of angiosperms [79,80]. We constrained our molecular clock analysis so that the Southern Hemisphere Macropathinae shared a common ancestor with Asiatic Aemodogryllinae and Rhaphidophorinae about 160 Mya, which agrees with a hypothesis based on morphology [12,13].

Divergence times inferred from our alternative analysis using secondary landscape calibrations were significantly younger than those estimated with our fossil calibration. Inconsistency in estimates derived from geophysical models of landscape changes have previously been observed [7,32], and ambiguity about the timing and biological influence of paleoenvironments such as Beringia (e.g., [81,82]) suggests that they are probably inappropriate for constraining phylogeny. The spatial distribution of lineage diversity in the Macropathinae also shows that the location of extant representatives of deep lineages does not indicate the ancestral location [26,76,83].

We inferred that a common ancestor of the two Asian subfamilies Aemodogryllinae and Rhaphidophorinae existed during the Late Jurassic/Early Cretaceous (155–121 Mya), when Gondwana and Laurasia were breaking up, and prior to the Late Cretaceous (110–65 Mya) Beringia land connection of North America and Asia. Moreover, stratigraphic evidence indicates that subduction of the Indian tectonic plate under Asia began about 61 Mya [84], with the subsequent formation of connecting land masses through continental collision about 45 Mya [85]. In our analysis, the earliest diversification of representative Aemodogryllinae and Rhaphidophorinae from Bhutan, Southeast China, and Japan occurred before the Tertiary contact of the Indian subcontinent and Eurasian tectonic plates [86]. The estimated diversification of *Diestramima* starting at about 63 Mya is inconsistent with the timing of the Himalayan emergence and the associated change in weather patterns thought to have influenced the diversification of this genus in Southeast China [16].

Our sample of *Stonychophora* from the Solomon Islands is at the southernmost end of the known range of the Rhaphidophorinae and brings this subfamily geographically close to the Macropathinae that span the Southern Hemisphere (Figure 1). In the current analysis *Stonychophora* is sister to *Rhaphidophora* species sampled from Bhutan, and their common ancestor was estimated to have existed about 106 Mya. We know through radiometric dating that the Solomon Islands archipelago started to form from volcanic activity 45–40 Mya [87], and Vangunu Island, where the specimen was collected, formed from more recent Plio-Pleistocene volcanics [88]. This indicates a complex biogeographic history of *Stonychophora* in the archipelago, Papua New Guinea, and through the Malay Archipelago—a region long recognized as a biogeographic nexus [89]. The establishment of such island biotas requires colonization from dispersal over water [6,8,90], which could obscure any signal of the spatial origins of subfamily lineages. The Solomon Islands archipelago spans a large area, with islands of various ages, so further sampling in this region is needed to interpret rhaphidophorid diversity in the region and its evolutionary history in relation to other Rhaphidophorinae in Southeast Asia. Nevertheless, it appears that the ancestral connection of these lineages to the Asian continent was via Southeast Asian, through the Western Pacific, rather than through India or Beringia.

## Figures and Tables

**Figure 1 insects-16-00670-f001:**
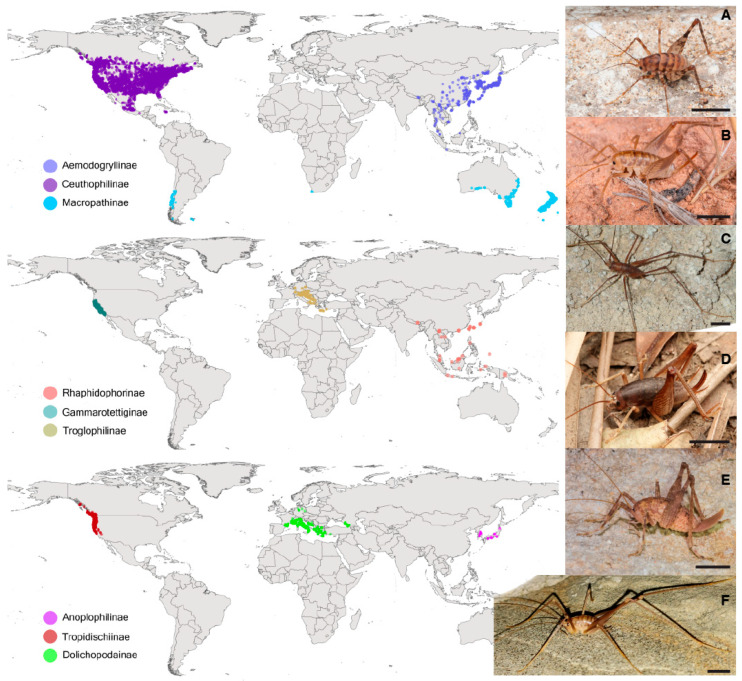
Global distribution of rhaphidophorid subfamilies based on 23,862 locality records retrieved from iNaturalist, GBIF (https://www.gbif.org/) and BOLD System (https://v3.boldsystems.org/). Data processed in QGIS v3.22 and edited in Inkscape v1.2.1. The subfamily Tropidischiinae comprises a single species (*Tropidischia xanthostoma*) of uncertain affinity. The invasive range of the greenhouse rhaphidophorid *Tachycines asynamorus* in North America and Europe is excluded. (**A**) Aemodogryllinae: male *Tachycines asynamorus*, China (Zhangyiyan). (**B**) Ceuthophilinae: female *Ceuthophilus* sp., USA (Danilo Hegg). (**C**) Dolichopodainae: female *Dolichopoda* sp., Itay (Fabrizio Mujica). (**D**) Rhaphidophorinae: female *Rhaphidophora taiwana* Okinawa, Japan (Orthoptera-JP). (**E**) Troglophilinae: female *Troglophilus cavicola*, Italy (Marco Vicariotto). (**F**) Macropathinae: male *Macropathus filifer*, NZ.

**Figure 2 insects-16-00670-f002:**
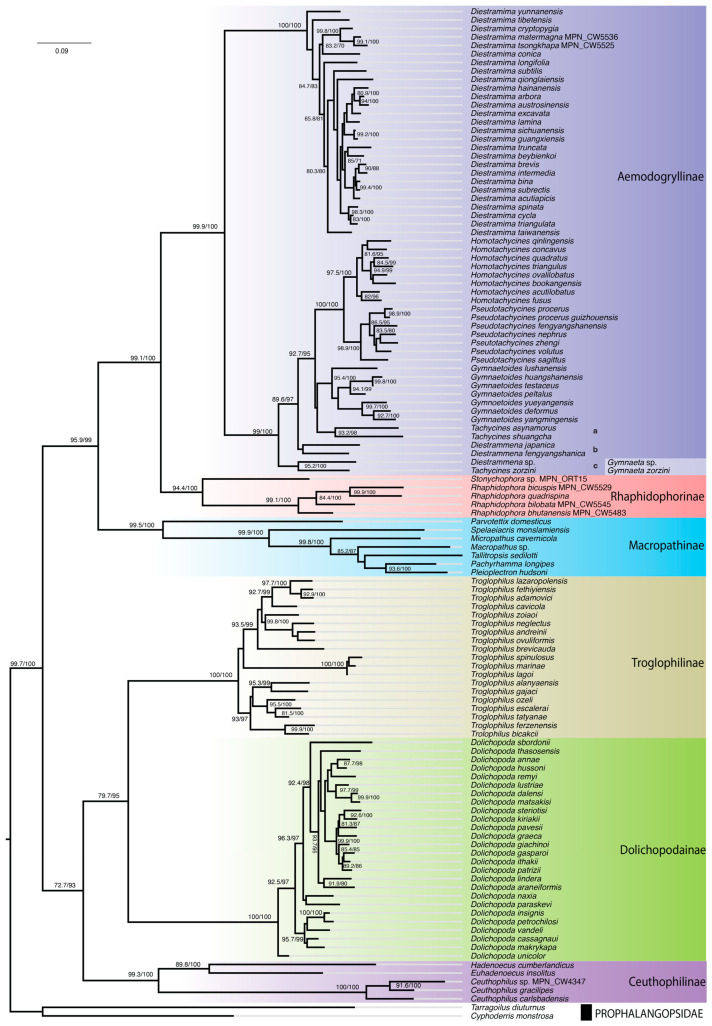
Phylogeny of camel crickets (Rhaphidophoridae) inferred from DNA sequences. ML analysis based on concatenated alignment of partial mitochondrial (COI, 12S, and 16S) and nuclear rRNA (18S and 28S) gene sequences (total: 3314 bp). ML bootstrap proportions > 90% of 1000 replicates are shown at nodes along with respective Sh-aLRT values. Labels a, b, and c in the tree (in subfamily Aemodogryllinae) indicate three monophyletic lineages comprising the paraphyletic genera *Diestrammena* and *Tachycines*. Resurrection of *Gymnaeta* for lineage c would render lineages a (*Tachycines*) and b (*Diestrammena*) monophyletic.

**Figure 3 insects-16-00670-f003:**
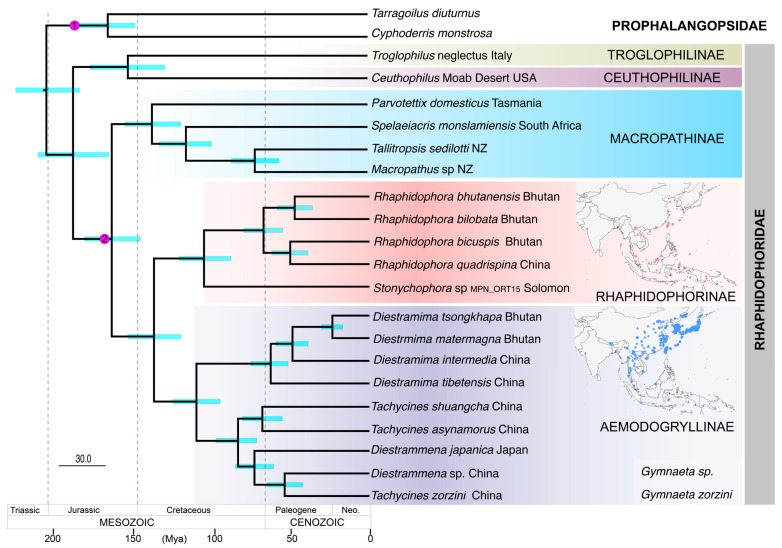
Chronogram of Rhaphidophoridae focusing on Asian lineages inferred by Bayesian analysis of 13 protein-coding genes calibrated with fossil and secondary fossil calibration points and a relaxed molecular clock. Calibration (1) fossil *Aboilus* (Prophalangopsidae) at approximately 172 Mya and (2) secondary fossil calibration point at 160 Mya as node ages of Macropathinae and Aemodogryllinae + Rhaphidophorinae (Table 2). Turquoise bars at nodes indicate 95% HPD credibility intervals. Inset maps indicate known locality records for the two subfamilies Rhaphidophorinae and Aemodogryllinae.

## Data Availability

The original data presented in the study are openly available in FigShare at 10.6084/m9.figshare.29396369 or https://evolves.massey.ac.nz/Data.htm. Additionally, all the newly sequenced mitochondrial genomes can be accessed from GenBank https://www.ncbi.nlm.nih.gov/genbank/.

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
