# Peer review of "Cretaceous Connections Among Camel Cricket Lineages in the Himalaya Revealed Through Fossil-Calibrated Mitogenomic Phylogenetics"

_insects, 2025, doi:10.3390/insects16070670_

Round 1
Reviewer 1 Report
Comments and Suggestions for Authors
This paper is considered to be of particular value in the field of orthopterans. It boldly challenges previous hypotheses and proposes a new perspective by incorporating data from specimens obtained from the Himalayas, a noteworthy and interesting region. The biogeographical information included in this paper, which incorporates these continental drift hypotheses, including estimates of divergence dates, will be of great interest to readers. The findings have the potential to inform future research on the biogeographical information of insects between eastern Asia and North America and related topics. I believe that this paper is worthy of publication.
The methodology is sound, and the data are analyzed rigorously. The results are novel and contribute meaningfully to the current literature. The main text is well-written and logically structured.
However, the supplement files still require some revisions.
For example, not all sample collection location data includes country names, and some typographical errors need to be corrected.
I recommend it for publication with only minor revisions.

Author Response
This paper is considered to be of particular value in the field of orthopterans. It boldly challenges previous hypotheses and proposes a new perspective by incorporating data from specimens obtained from the Himalayas, a noteworthy and interesting region. The biogeographical information included in this paper, which incorporates these continental drift hypotheses, including estimates of divergence dates, will be of great interest to readers. The findings have the potential to inform future research on the biogeographical information of insects between eastern Asia and North America and related topics. I believe that this paper is worthy of publication.
The methodology is sound, and the data are analyzed rigorously. The results are novel and contribute meaningfully to the current literature. The main text is well-written and logically structured. However, the supplement files still require some revisions. For example, not all sample collection location data includes country names, and some typographical errors need to be corrected. I recommend it for publication with only minor revisions.
>>Thank you for your positive and valuable comments.
Specific comment
However, the supplement files still require some revisions.
>> Done.
For example, not all sample collection location data includes country names, and some typographical errors need to be corrected.
>> Corrected all typographical errors and included country names for all sample collection data (supplementary file attached).
Reviewer 2 Report
Comments and Suggestions for Authors
Simple Summary and Abstract
Different information is found in these summaries. The two together make a coherent Abstract. The former is presumably to be the same as the latter with simplified terminology and findings.
These sections will be rewritten once the goals of the study have been decided upon and clearly communicated in the Introduction.
Is the purpose of the paper testing global paleogeographic hypotheses? Testing phylogeographic hypotheses relevant to Asia? Genus-level systematics? Increasing regional sampling of Rhaphidophoridae?
My impression is that the dataset is best suited to testing specific Eurasian phylogeographic hypotheses, given that several regional subfamilies are missing. For a take home message in the Simple Summary, communicate the significance to the study of biogeography as a whole, justifying why increased sampling of Asian fauna helps with those goals.
I have extractions of several Gammarotettiginae if the authors are interested! I can probably also get Tropidischiinae.
Introduction
Needs restructuring. State coherent goal(s) or question(s) at the beginning of the Intro before organism background. Lead with the phylogeographic hypotheses or questions if that is the purpose of the paper. Then introduce the Rhaphidophoridae and explain what they may tell us about phylogeography. Next go into taxonomy, biology, and dispersal potential. If phylogeography is the goal, what is the benefit to the field of increased understanding of Asian Rhaphidophoridae, especially if the leading hypothesis of subfamily diversification is Pangaea breakup? Summarize applicable phylogeographic hypotheses or scenarios, especially if they are to be referred to in the Discussion.
Topics are scattered among paragraphs. Bringing topic information together into one or two paragraphs each will help the Intro read more coherently (see specific comments below).
Rhaphidophoridae are definitely understudied among families of Orthoptera and deserve attention on this merit alone.
Introduce conserved morphology and give examples. This justifies a molecular systematics approach. A goal of taxonomic revision is not set in the Intro, yet the Results and Discussion begin with some taxonomic rearrangements that lack accompanying morphological reinterpretation or redescription. Taxonomy thus seems like a minor component of this work that may be better left to forthcoming papers that test the congruence of several character datasets. Is there enough evidence to warrant the proposed rearrangements?
Specific comments:
pg. 1. mixes several topics: diversity, habitus, general biology, and dispersal. No main goals or questions are communicated except that Rhaphidophoridae could use more study, which is definitely true.
pg. 3 line 68. Paragraph unclear. Circular reasoning of paleogeographic methodology is mentioned but if the point is moot with any calibration methods scarce for Rhaphidophoridae, why mention it? This mention becomes problematic as the work employs some circular methods and may not satisfy the requirement of total evidence (see Methods and Results comments below). This Intro component may preface a solution the authors came up with for calibration, which is not explained until Methods. If so, suggest potential solutions here.
pg. 5. Topic is phylogenetic position of Rhaphidophoridae. Makes sense to put this information earlier in the Intro after the goals and near relevant phylogeographic hypotheses. An ancient group like this has strong potential to contribute to phylogeography. The pattern of amphitropical distributions, apparent from Fig. 1, is characteristic of ancient groups, and is perhaps worth pointing out.
pg. 6. Purpose of study. Sampling only 5 of 9 subfamilies reduces the ability of the work to draw conclusions about global phylogeography if that is the goal. Perhaps restrict the goal to testing phylogeography pertinent to Eurasia. Lay out potential hypotheses for this region specifically and the evidence required to support them.
line 60. Gammarotettiginae misspelled.
Fig. 1 caption (F) missing, does it belong to the macropathine?
Methods
How suitable is mitogenomic data to resolve questions of origin in an ancient lineage? Mitochondrial genes have generally fast mutation rates. Is saturation a potential problem with an ancient lineage? Consider including a test for substitution saturation at relevant protein coding loci.
Furthermore, a mitochondrion amounts to a single locus for purposes of linked inheritance. Mitochondrial phylogenomics may produce strong support for erroneous relationships without including other unlinked loci. The authors may caution about some of these pitfalls in the Intro. The 117 taxon dataset may be more informative despite containing a smaller number of characters, as it benefits from both larger taxon and locus sampling (two linked loci: nuclear rDNA and all mitochondrial genes). It would be interesting to run BEAST for the 117 taxon dataset with a two-partition model for the two linkage groups and compare with the mtgenomic dataset. Regardless, this work remains important despite those pitfalls given the understudied nature of the Rhaphidophoridae, if that is a primary stated goal.
line 171. What happened to nuclear histone sequences? Were they analyzed further in any dataset?
Standard mtgenomics methods and BEAST analyses well explained for the final analysis. From the Results, however, preliminary/exploratory analyses are mentioned that are not explained. If the authors used preliminary analyses to derive priors for the final analyses, this is an instance of circular reasoning other than the kind cautioned in the Intro.
Upon further thought, this work may not adhere to the requirement of total evidence. Rather than run separate analyses and compare them, a Bayesian framework provides the tools necessary to compete hypothesis with a single analysis. From the Intro, the authors put more stock in fossil calibrations than landscape processes. These may form weighted priors, which the ensuing analysis will apportion among the posterior probability.
Fig. 1 is a great map. What methods were used to generate it once the GIS data were obtained? Are there really no Rhaphidophoridae in Africa?
Results
Begins with some genus-level systematics. No systematics goals were stated in the Intro. This component seems premature. Diestrammena et al. are large complexes of genera that would require a large scale revision. If genera are being resurrected or elevated from subgenera, morphological evidence is warranted.
Systematics and phylogeography are clearly reported. Congruent results make for straightforward communication of Results.
line 317. Reports exploratory analyses not explained in Methods. As the trial analyses found priors for final analyses, those Methods must be reported before the Results. The use of preliminary analyses to derive priors for final analyses may be circular. Perhaps results are not as congruent as reported, given selective choice of priors and bias towards certain phylogeographic hypotheses.
Fig. 3 caption will properly read “relaxed” molecular clock on line 358.
Fossil calibration and geographic calibration give much different estimates. A single analysis that includes both scenarios as priors may help draw conclusions about the support for each given the resulting posterior.
Discussion
The taxonomic revisions come out of nowhere: such goals were not mentioned in the Intro and do not have any supporting Results beyond topology. Genera are revised in this work without accompanying morphological reinterpretation or redescription. Minimal morphological characters are mentioned in this section but are not mentioned in the Intro. It seems premature to rearrange genus and species-level taxonomy at this time without an attempt at morphological analysis. Perhaps a small section and a table in Results will suffice to summarize the morphological implications for these rearrangements.
Use the criterion of total evidence. A feature of Bayesian methods is weighting of prior information in the form of existing hypotheses and researcher biases. The fossil and landscape-derived priors both have uncertainty and the researchers introduce their own biases into which dataset is more reliable. A purpose of Bayesian statistics is to sort this all out into measurable posterior weights.
There are numerous paleogeographical events that receive first mention in the Discussion. Introduce important events in the Introduction. Some of these may form priors for Bayesian analysis.
When the authors choose a set of goals for the study, the Discussion may be rearranged to address each of those goals in turn.
Literature Cited
Several genera not italicized in references.
Author Response
REVIEWER 2
Simple Summary and Abstract
Different information is found in these summaries. The two together make a coherent Abstract. The former is presumably to be the same as the latter with simplified terminology and findings.
These sections will be rewritten once the goals of the study have been decided upon and clearly communicated in the Introduction.
>>Made changes to summary and abstract considering the feedback.
Is the purpose of the paper testing global paleogeographic hypotheses? Testing phylogeographic hypotheses relevant to Asia? Genus-level systematics? Increasing regional sampling of Rhaphidophoridae? My impression is that the dataset is best suited to testing specific Eurasian phylogeographic hypotheses, given that several regional subfamilies are missing. For a take home message in the Simple Summary, communicate the significance to the study of biogeography as a whole, justifying why increased sampling of Asian fauna helps with those goals.
>>Thanks for these suggestions. We have endeavoured to clarify our intension by reorganising some of the material in the Introduction section.
I have extractions of several Gammarotettiginae if the authors are interested! I can probably also get Tropidischiinae.
>> Nice, Thanks!
Introduction
Needs restructuring. State coherent goal(s) or question(s) at the beginning of the Intro before organism background. Lead with the phylogeographic hypotheses or questions if that is the purpose of the paper. Then introduce the Rhaphidophoridae and explain what they may tell us about phylogeography. Next go into taxonomy, biology, and dispersal potential. If phylogeography is the goal, what is the benefit to the field of increased understanding of Asian Rhaphidophoridae, especially if the leading hypothesis of subfamily diversification is Pangaea breakup? Summarize applicable phylogeographic hypotheses or scenarios, especially if they are to be referred to in the Discussion.
>> (General comment about our Introduction flow)
We think that our introduction follows a standard flow where we have tried to give a general overview/background about the insect (habit and habitats). Followed by detail of its distribution across different regions and what it implies in phylogeographic study. Our third paragraph basically deals with the phylogenetic positions of the family and its associated phylogeography hypothesis (based on past study). Fourth and fifth paragraphs deals with past phylogeogphic studies and methods used for calibrations. And finally with a background on taxonomy of problematic taxa belonging to subfamily Aemodogryllinae and ending with a paragraph of summary our study.
Topics are scattered among paragraphs. Bringing topic information together into one or two paragraphs each will help the Intro read more coherently (see specific comments below). Rhaphidophoridae are definitely understudied among families of Orthoptera and deserve attention on this merit alone.
>> Thanks, we have done some restructuring, particularly we have reorganised paragraph 3 and 4 [lines 71–87 and 98–103].
Introduce conserved morphology and give examples. This justifies a molecular systematics approach. A goal of taxonomic revision is not set in the Intro, yet the Results and Discussion begin with some taxonomic rearrangements that lack accompanying morphological reinterpretation or redescription. Taxonomy thus seems like a minor component of this work that may be better left to forthcoming papers that test the congruence of several character datasets. Is there enough evidence to warrant the proposed rearrangements?
>>Part of systematics goal is included in introduction [lines 116–130]. We are not suggesting major taxonomic changes but recommending the use of a genus name that has been neglected for a few years. This suggestion currently effects only two species and therefore does not require morphological reinterpretation or redescription.
Specific comment
- 1. mixes several topics: diversity, habitus, general biology, and dispersal. No main goals or questions are communicated except that Rhaphidophoridae could use more study, which is definitely true.
>> This section is an introduction to our study insect. The first paragraph gives a general overview about the habit and habitat which we thought would serve a perfect background for readers.
- 3 line 68. Paragraph unclear. Circular reasoning of paleogeographic methodology is mentioned but if the point is moot with any calibration methods scarce for Rhaphidophoridae, why mention it? This mention becomes problematic as the work employs some circular methods and may not satisfy the requirement of total evidence (see Methods and Results comments below). This Intro component may preface a solution the authors came up with for calibration, which is not explained until Methods. If so, suggest potential solutions here.
>>In this paragraph we cover the literature about phylogenetic analyses (and molecular clocks) which use many different approaches, some of which are known to be circular in nature. In our methods section we point out the details of the two approaches we have used. We are explicit that using paleogeographic calibrations are intended as a comparison with the preferred fossil calibration approach. Combining these two methods (“total evidence”) into a single molecular clock analysis is not possible as the lack of compatibility results in failure of the model to converge.
- 5. Topic is phylogenetic position of Rhaphidophoridae. Makes sense to put this information earlier in the Intro after the goals and near relevant phylogeographic hypotheses. An ancient group like this has strong potential to contribute to phylogeography. The pattern of amphitropical distributions, apparent from Fig. 1, is characteristic of ancient groups, and is perhaps worth pointing out.
>>We think that our introduction follows a standard flow where we have given a general overview about the insect (habit and habitats), followed by details of its distribution across different regions and what it implies in phylogeographic study. Our third paragraph deals with the phylogenetic positions of the family and its associated phylogeography hypothesis (based on past study). Fourth and fifth paragraphs deals with past phylogeographic studies and methods used for calibrations. And finally with a background on taxonomy of problematic taxa belonging to subfamily Aemodogryllinae and ending with a paragraph summarising our study.
- 6. Purpose of study. Sampling only 5 of 9 subfamilies reduces the ability of the work to draw conclusions about global phylogeography if that is the goal. Perhaps restrict the goal to testing phylogeography pertinent to Eurasia. Lay out potential hypotheses for this region specifically and the evidence required to support them.
>> See our response above. These five subfamilies of Rhaphidophoridae represent all biogeographic regions.
line 60. Gammarotettiginae misspelled.
>>Corrected (highlighted)
Fig. 1 caption (F) missing, does it belong to the macropathine?
>>Yes. Corrected (highlighted)
Methods
How suitable is mitogenomic data to resolve questions of origin in an ancient lineage? Mitochondrial genes have generally fast mutation rates. Is saturation a potential problem with an ancient lineage? Consider including a test for substitution saturation at relevant protein coding loci.
>>Mitogenomic data has been used to infer phylogenetic trees for many insect lineages (for example cockroaches Bourguignon et al. 2018 Mol. Biol. Evol. 35(4):970–983 doi:10.1093/molbev/msy013; Grasshoppers Koot et al. 2020 https://doi.org/10.1016/j.ympev.2020.106783). Partition modelling within our phylogenetic analysis allows for differences in rate and signal capacity.
Furthermore, a mitochondrion amounts to a single locus for purposes of linked inheritance. Mitochondrial phylogenomics may produce strong support for erroneous relationships without including other unlinked loci. The authors may caution about some of these pitfalls in the Intro. The 117 taxon dataset may be more informative despite containing a smaller number of characters, as it benefits from both larger taxon and locus sampling (two linked loci: nuclear rDNA and all mitochondrial genes). It would be interesting to run BEAST for the 117 taxon dataset with a two-partition model for the two linkage groups and compare with the mtgenomic dataset. Regardless, this work remains important despite those pitfalls given the understudied nature of the Rhaphidophoridae, if that is a primary stated goal.
>>Yes, we agree that as the mitochondria is a single locus caution is needed in case of hybridisation/introgression misleads interpretation. When working with closely related species a gene tree might not represent the species tree. However, we are inferring phylogenetic relationships from very distantly related taxa and therefore introgression is very unlikely to be a problem. This inference is corroborated by our analysis of 117 taxon dataset which integrated available short sequences from mitochondrial COI, 16S, 12S and nuclear rRNA 28S and 18S.
line 171. What happened to nuclear histone sequences? Were they analyzed further in any dataset?
>> Our histone data is novel and homologous data representing relevant taxa are not currently available from other studies. Thus, at this stage we are unable to include these data in the present phylogenetic analyses.
Standard mtgenomics methods and BEAST analyses well explained for the final analysis. From the Results, however, preliminary/exploratory analyses are mentioned that are not explained. If the authors used preliminary analyses to derive priors for the final analyses, this is an instance of circular reasoning other than the kind cautioned in the Intro.
>> No, preliminary analyses were not used to derive priors, they were used to determine whether models could converge given the combination of calibrations and distributions tried. This exploratory approach is standard practice (e.g. Nowak et al. 2013 https://doi.org/10.1371/journal.pone.0066245).
Upon further thought, this work may not adhere to the requirement of total evidence. Rather than run separate analyses and compare them, a Bayesian framework provides the tools necessary to compete hypothesis with a single analysis. From the Intro, the authors put more stock in fossil calibrations than landscape processes. These may form weighted priors, which the ensuing analysis will apportion among the posterior probability.
>>We have not suggested that we viewed “Total evidence” as an important requirement within this study. Indeed, the idea that any evolutionary study has total evidence is false. Combining fossils and biogeographic constraints into a single molecular clock analysis is not possible as they conflict and prevent the model from converging. With different calibration methods it would not be possible to compare goodness-of-fit. Our study did contrast inferred date estimates from different approaches allowing discussion of the merits of fossil calibration. Fossil calibration for this type of study is preferred because fossils deal with taxonomic lineage formation whereas palaeogeographic calibration makes the circular assumption that land changes cause lineage formation.
Fig. 1 is a great map. What methods were used to generate it once the GIS data were obtained? Are there really no Rhaphidophoridae in Africa?
>> Thanks. Methods used are now provided in brief with Fig. 1 caption. Macropathinae species are found in south of Africa which can be seen in top map (quite hard to see but in blue).
Results
Begins with some genus-level systematics. No systematics goals were stated in the Intro. This component seems premature. Diestrammena et al. are large complexes of genera that would require a large scale revision. If genera are being resurrected or elevated from subgenera, morphological evidence is warranted.
>> Thank you. We have added a paragraph in the introduction to explain.
Our comments are based on the well-recognised expectation of monophyly, so we still recommend resurrecting the genus. However, considering the feedback, morphological evidence is provided here: Adelung, N.von (1902) Beitrag zur Kenntnis der Paläarctischen Stenopelmatiden (Orthoptera, Locustodea). Annuaire du Musée Zoologique de l'Académie Impériale des Sciences de St.-Pétersbourg, 7, 55–75.
Systematics and phylogeography are clearly reported. Congruent results make for straightforward communication of Results.
>> Thanks.
line 317. Reports exploratory analyses not explained in Methods. As the trial analyses found priors for final analyses, those Methods must be reported before the Results. The use of preliminary analyses to derive priors for final analyses may be circular. Perhaps results are not as congruent as reported, given selective choice of priors and bias towards certain phylogeographic hypotheses.
>> Yes, we agreed our methods were not explicit, and have now reported in the last paragraph of heading ‘Divergence time estimate analysis’ under method section.
The method is not circular. This is a standard method to evaluate convergence and EES scores for applying different priors that allow us to use best prior combinations for final analysis. As explained in our result, since all the analysis started to converge between 1–4 million generations with stable node ages and considerable good ESS (250) for all the priors (after removing 10-40% as burn-in), there will be no considerable changes in the final estimates.
Fig. 3 caption will properly read “relaxed” molecular clock on line 358.
>>Agreed and corrected (highlighted)
Fossil calibration and geographic calibration give much different estimates. A single analysis that includes both scenarios as priors may help draw conclusions about the support for each given the resulting posterior.
>>That would be illogical as they involve contrary assumptions.
Discussion
The taxonomic revisions come out of nowhere: such goals were not mentioned in the Intro and do not have any supporting Results beyond topology. Genera are revised in this work without accompanying morphological reinterpretation or redescription. Minimal morphological characters are mentioned in this section but are not mentioned in the Intro. It seems premature to rearrange genus and species-level taxonomy at this time without an attempt at morphological analysis. Perhaps a small section and a table in Results will suffice to summarize the morphological implications for these rearrangements.
>> We have now indicated briefly about the taxonomy study in the introduction (6th paragraph). We fully agree with the reviewer regarding the taxonomic changes, and we lack specimens to fully support the genus and species-level taxonomy. However, considering our genetic analysis, we have suggested future detail systematic study on this highly disputed group of Asian lineages as indicated in first paragraph of our discussion. We have removed the “comb. nov.” from our phylogenetic tree and discussion.
Use the criterion of total evidence. A feature of Bayesian methods is weighting of prior information in the form of existing hypotheses and researcher biases. The fossil and landscape-derived priors both have uncertainty and the researchers introduce their own biases into which dataset is more reliable. A purpose of Bayesian statistics is to sort this all out into measurable posterior weights.
>>Combining fossils and biogeographic constraints into a single molecular clock analysis is not possible as the lack of compatibility results in failure of the model to converge.
There are numerous paleogeographical events that receive first mention in the Discussion. Introduce important events in the Introduction. Some of these may form priors for Bayesian analysis. When the authors choose a set of goals for the study, the Discussion may be rearranged to address each of those goals in turn.
>>Thanks, we have made small adjustments to our introduction and discussion so that the goals of this study are clearly addressed
Literature Cited
Several genera not italicized in references.
>>Edited (highlighted)
Reviewer 3 Report
Comments and Suggestions for Authors
This is a groundbreaking study that adds new Rhaphidophorinae sequence data from Bhutan and utilizes not only secondary calibration but also fossil record calibration to re-evaluate the divergence time estimates among subfamilies of Rhaphidophoridae as presented in Kim et al. (2004). The proposed divergence times offer a new perspective, making this study highly valuable for publication.
In Kim et al. (2004), the estimated divergence times among subfamilies were largely consistent with the timing of major geographical events, making their results easy for readers to understand. In contrast, this study suggests that the divergence among subfamilies occurred earlier than the estimates in Kim et al. (2004). However, it does not explicitly present hypotheses regarding the events that facilitated subfamily diversification. This lack of clarification makes the study more complicated to interpret. Ideally, it would be beneficial to estimate diversification events beyond vicariant speciation, though this may be challenging.
Therefore, the use of fossil record calibration in this study raises the question: why has the estimated divergence time changed? Furthermore, it would strengthen the manuscript to explicitly discuss how the divergence time estimates differ from previous research and, in the authors' assessment, why their new estimates are more reliable. Clarifying these points in the Discussion section would improve the paper.
Additionally, there were two instances in the manuscript where “ESS value” and “ESS statistics” were mistakenly written as "EES." These should be corrected.
Author Response
REVIEWER 3
This is a groundbreaking study that adds new Rhaphidophorinae sequence data from Bhutan and utilizes not only secondary calibration but also fossil record calibration to re-evaluate the divergence time estimates among subfamilies of Rhaphidophoridae as presented in Kim et al. (2004). The proposed divergence times offer a new perspective, making this study highly valuable for publication.
>>Thank you for your positive and valuable comments.
In Kim et al. (2004), the estimated divergence times among subfamilies were largely consistent with the timing of major geographical events, making their results easy for readers to understand. In contrast, this study suggests that the divergence among subfamilies occurred earlier than the estimates in Kim et al. (2004). However, it does not explicitly present hypotheses regarding the events that facilitated subfamily diversification. This lack of clarification makes the study more complicated to interpret. Ideally, it would be beneficial to estimate diversification events beyond vicariant speciation, though this may be challenging.
Therefore, the use of fossil record calibration in this study raises the question: why has the estimated divergence time changed? Furthermore, it would strengthen the manuscript to explicitly discuss how the divergence time estimates differ from previous research and, in the authors' assessment, why their new estimates are more reliable. Clarifying these points in the Discussion section would improve the paper.
>>We agree with the reviewer-3 that our study suggests divergence times that are not consistent with simple geographical events as illustrated by Kim et al. 2024. Just because our date estimates do not coincide with geological events makes them harder to explain but does not invalidate them. We think a more detail phylogeographic interpretation is beyond the scope of our evidence and our paper. However, we have briefly mentioned in our introduction about the different calibration used in past studies (line 105–115) and what we used in our method sections (line 252–264). Our restructuring of the material in the introduction should make the focus clear and given our lack of a simple explanation of diversification we have not expanded our discussion much beyond the evidence.
Additionally, there were two instances in the manuscript where “ESS value” and “ESS statistics” were mistakenly written as "EES." These should be corrected.
>> Thank you, now corrected (highlighted)
Reviewer 4 Report
Comments and Suggestions for Authors
This manuscript presents a well-written and structured investigation into the evolutionary relationships and biogeographic history of flightless camel crickets (Rhaphidophoridae), with a focus on Asian lineages. The authors have carefully employed molecular clock analyses to estimate divergence times of major lineages. The study provides important evidence, particularly regarding taxonomic inconsistencies and divergence periods. While some critical points require further verification in future analyses, the analytical process has been conducted with care.
Specific Comments:
L68-69: Are the quotation marks around the phrase “since it is founded on an assumption of the very question to be decided” necessary?
L168: Please add a space between '11,124' and 'nucleotides': '11,124 nucleotides'.
L310: For clarity, please specify 'gene concordance factor' instead of just 'concordance factor'.
L394-396: Please rephrase this sentence for clarity. I am having difficulty understanding the intended meaning regarding prior distribution models significantly impacting the estimates.
L401: Please remove the space between '{71}' and the period.
L444-446: The connection between the diversification of Asian subfamilies and the Beringia land bridge is not clear. Please elaborate on how Beringia is relevant given the estimated divergence times.
Author Response
REVIEWER 4
Specific comments
L68-69: Are the quotation marks around the phrase “since it is founded on an assumption of the very question to be decided” necessary?
>>It is the exact phrase directly from Wallace, 1876 paper so we find it is necessary.
L168: Please add a space between '11,124' and 'nucleotides': '11,124 nucleotides'.
>> Corrected (highlighted)
L310: For clarity, please specify 'gene concordance factor' instead of just 'concordance factor'.
>>Agreed and corrected (highlighted).
L394-396: Please rephrase this sentence for clarity. I am having difficulty understanding the intended meaning regarding prior distribution models significantly impacting the estimates.
>>It meant use of different prior setting in testing our calibration has no significant impact on estimations of common ancestors as shown in table 2. So, therefore, now reads “We showed that the prior distribution models (normal, log-normal or exponential) used for our calibrations of molecular clock analyses did not significantly impact our estimates of the timing of common ancestors” (line 419–424).
L401: Please remove the space between '{71}' and the period.
>>Corrected (highlighted).
L444-446: The connection between the diversification of Asian subfamilies and the Beringia land bridge is not clear. Please elaborate on how Beringia is relevant given the estimated divergence times.
>>Here proposed to refute the role of Beringia for the diversification of Asian subfamilies based on our estimates.
Round 2
Reviewer 2 Report
Comments and Suggestions for Authors
Introduction is much clearer, thanks. The reordering focuses the goals, frames biogeographic hypotheses, and explains the systematic importance of the taxonomic revisions.
Methods
line 242 Prophalangopsidae misspelled.
The histone sequences still do not appear in any phylogenetic dataset.
Explanation of exploratory methods is satisfactory.
Discussion includes improved explanation of hypothesis testing. The authors’ interpretations are clearly stated as hypotheses with suggestions for further research.
The phylogeographic discussion is interesting, and considers conflict among datasets.
Author Response
Introduction is much clearer, thanks. The reordering focuses the goals, frames biogeographic hypotheses, and explains the systematic importance of the taxonomic revisions. Explanation of exploratory methods is satisfactory. Discussion includes improved explanation of hypothesis testing. The authors’ interpretations are clearly stated as hypotheses with suggestions for further research. The phylogeographic discussion is interesting and considers conflict among datasets.
>>Thank you for your constructive feedback in improving our manuscript.
line 242 Prophalangopsidae misspelled.
>> Corrected.
The histone sequences still do not appear in any phylogenetic dataset.
>> This information has been removed from both the Methods section and the 'Data Availability' section of the supplementary documents.